# China's Three Major Cereal Crops Exposure to Compound Drought and Extreme Rainfall Events

**Hanming Cao[a] \*,  Qiren Yang[a]\*,  Wei Yang[a]\*,  Lin Zhao[a#]**

[a] School of Resource and Environmental Sciences, Wuhan University, Wuhan 430079, China

Correspondence to:  Lin Zhao (linzhao@whu.edu.cn)

**Abstract**

Under the backdrop of global climate change, the increasing intensity and frequency of anomaly climate events have led to a rise in compound extreme events. China's large population exacerbates the pressure of agricultural production, and compound drought and extreme rainfall events (CDER) can cause considerable damage to soil structure, thereby disrupting normal agricultural activities. Previous studies have revealed the impacts of the individual event, but the spatiotemporal characteristics of CDER and their effects on agricultural production remain obscure. This study focuses on compound disaster events in China's nine major agricultural regions, where drought and extreme rainfall events occur within 5 days. The results show that compound disasters are mainly concentrated in the northwest, southwest, and northern regions. The impact area of compound disasters is largest in summer, and the frequency and intensity of drought-rainfall events are higher than those of rainfall-drought events. Further analysis at the crop growth stage scale reveals the exposure of the three major cereal crops (rice, wheat, and maize) during their growth stage. The study reveals that maize generally has the highest and most variable disaster risk, rice has the lowest risk with minimal fluctuations, and wheat has moderate risk with large variations. The risk evolution in each agricultural region follows a universal pattern of "first rising and then declining", with the peak occurring around 2010. This study elucidates the spatiotemporal distribution patterns of this novel compound disaster and provides constructive insights for disaster prevention and mitigation through more refined risk assessments.

Keywords: compound drought and extreme rainfall events; crop maturity exposure; nine major agricultural regions in China; spatiotemporal distribution characteristics

*These authors contributed equally.
#Correspondence to: Lin Zhao (linzhao@whu.edu.cn)

## 1. Introduction

Climate change is one of the most serious threats to human society in the 21st century and may generate more extreme weather events and show an increasing trend at regional and global scales under anthropogenic climate change (AghaKouchak et al., 2014; Leonard et al., 2014; Zscheischler et al., 2020). With the intensification of global warming, the frequency, intensity, and compound effects of extreme rainfall and drought events have shown significant increases (Fang et al., 2025; Hao et al., 2018; Walz et al., 2021). The Sixth Assessment Report of the IPCC indicates that the frequency of compound drought and extreme rainfall events globally increased by 34% between 1980 and 2020 compared to pre-industrial levels (Calvin et al., 2023). Farmers are constantly dealing with and managing various agricultural risks that may have compound effects (Van Winsen et al., 2013; Wauters et al., 2014). Compound extreme events can exacerbate the damage caused by individual events and push global socio-economic systems to tipping points (Dickinson et al., 2016). This is because the combined stressors can overwhelm the capacity of exposed natural and human systems to cope with extreme conditions (Jayaraman et al., 2025; Ruess et al., 2025). The hazards of compound drought and extreme rainfall events (CDER) are not only reflected in the individual effects of drought or extreme rainfall but also in their combined effects. These events can cause significant damage to soil structure: extreme rainfall-induced soil erosion leads to an annual loss of 240 billion tons of topsoil globally (Borrelli et al., 2017), while anomaly drought can reduce soil organic matter content by 40-60%. Extreme rainfall events, characterized by high intensity and short duration, can induce severe soil erosion through processes such as splash erosion and overland flow (Quansah, 1981; Wang et al., 2021). The resultant detachment and transport of soil particles not only degrade soil fertility but also contribute to sedimentation in water bodies, exacerbating water quality issues. This alarming rate of soil erosion underscores the vulnerability of agricultural lands and natural ecosystems to hydrological extremes. Conversely, prolonged drought conditions impose distinct yet equally detrimental impacts on soil structure (Vicente-Serrano et al., 2020). Droughts reduce soil moisture levels, which are critical for maintaining soil aggregate stability and porosity. The desiccation of soil organic matter (SOM) under drought stress leads to its accelerated decomposition, resulting in a significant reduction in SOM content

often by 40-60% (Goebel et al., 2011; Yang and Liu, 2020). This decline in SOM not only diminishes soil fertility but also compromises the soil's ability to retain water and nutrients, further exacerbating its susceptibility to erosion during subsequent rainfall events. Additionally, drought-induced changes in soil structure can reduce hydraulic conductivity, impairing water infiltration and increasing runoff generation during rainfall events, and the water-replenishing effect of rainfall is further limited (Caplan et al., 2019). The combined effect of these two factors can reduce soil productivity by up to 75% (Lal, 2015).

Recent studies indicate that compound extreme events will significantly increase globally, with the occurrence rate of related events potentially increasing by 20-40% by mid-century (Steensen et al., 2025). Since the 1990s, the frequency of "drought-flood abrupt alternation" events in the South China and Southwest regions has significantly increased by over 40% (Hui et al., 2013; Shen et al., 2012; Wang et al., 2009). These compound events pose a significant threat to China's agricultural production and ecological environment. Meanwhile, China, with only 7% of the world's arable land, must support 20% of the global population (Bongaarts, 2021). Moreover, the economic losses caused by these events are substantial and widespread. For example, in 2011, after experiencing continuous drought in winter, spring, and summer, the middle and lower reaches of the Yangtze River suffered from heavy rainfall, resulting in over 2 million hectares of affected cropland and direct economic losses of 29.36 billion yuan (Meteorological Publishing House, 2012). Over the past two decades, extreme rainfall has reduced China's rice production by one-twelfth (Fu et al., 2023). Against this backdrop, the urgency of enhancing agricultural climate resilience is highlighted.

Existing studies have focused on the phenomenon of drought and extreme rainfall alternation, but most compound event identification studies are limited to provincial scales rather than national scales (Barriopedro et al., 2011; Zhao et al., 2023). Some studies are motivated by the development of an integrated index to address the multidimensional nature of agricultural drought impacts, its spatial vulnerability perspective, and scale requirements (Murthy et al., 2015). Additionally, while some studies have quantified the risks of population and economic exposure to drought-flood abrupt alternation using shared socio-economic pathways (Meng et al., 2024), the exposure risks of directly affected crops remain unclear. Most studies have used hydrological indices to characterize compound events,

focusing on daily scales and using indices such as the standardized precipitation-evapotranspiration index. In contrast, this study uses soil moisture data to monitor compound events from an agricultural perspective rather than a hydrological one. Moreover, encountering compound events during critical crop phenological stages can amplify yield losses by 3-5 times (Lesk et al., 2022). The impact of compound events on crops at different growth stages is significant, but current agricultural studies have focused more on the exposure analysis of individual extreme rainfall and drought events, with limited research on compound event exposure. From a developmental perspective, existing studies have identified and analyzed drought-flood abrupt alternation events in China from daily, monthly, and annual scales, forming four important research hotspots and frontiers: identification methods, causation analysis, evolution characteristics, and disaster damages (Shen et al., 2018; Yang and Liu, 2020). However, there is still a lack of comprehensive, national-scale analyses of secondary CDER in agricultural regions. The spatiotemporal distribution and evolution characteristics of drought-flood abrupt alternation events in China remain unclear, and research on crop exposure in the nine major grain-producing regions is still a blank space.

We define CDER as disaster events where drought and extreme rainfall occur within a 5-day interval. Specifically, these can be divided into compound extreme rainfall-drought events $CDER_{rd}$ (extreme rainfall followed by drought within 5 days) and compound drought-rainfall $CDER_{dr}$ (drought followed by extreme rainfall within 5 days). The research statistically analyzes the frequency, intensity, monthly changes, and annual changes of these two types of events to reveal their spatiotemporal distribution characteristics comprehensively. Additionally, this study innovatively calculates the exposure of China's three major agricultural products (maize, wheat, and rice) during their maturation periods. By focusing on the crop growth stage, the study refines exposure calculations and trend analyses will aid in better disaster prevention and mitigation efforts based on an understanding of these compound disasters.

## 2.Materials and Methods

### 2.1 Study Area

China's agricultural regions are vast and geographically diverse, encompassing nine major agricultural zones that include plains, mountains, basins, and plateaus. These regions are crucial for grain production and span from 3°51′N to 53°33′N and 73°33′E to 135°05′E. The climate types are complex and varied, including tropical monsoon, subtropical monsoon, temperate monsoon, temperate continental, and alpine climates. Crop maturity systems range from one harvest per year to three harvests per year. The agricultural zoning data of China are derived from the China Agricultural Comprehensive Zoning Map released by the National Agricultural Commission (Fig. 1), where C1 represents the Northeast China Region; C2 represents the Inner Mongolia and Great Wall Contiguous Region; C3 represents the Gansu-Xinjiang Region; C4 represents the Huang-Huai-Hai Region; C5 represents the Loess Plateau Region; C6 represents the Qinghai-Tibet Region; C7 represents the Middle-Lower Yangtze River Region; C8 represents the Southwest China Region; and C9 represents the South China Region).

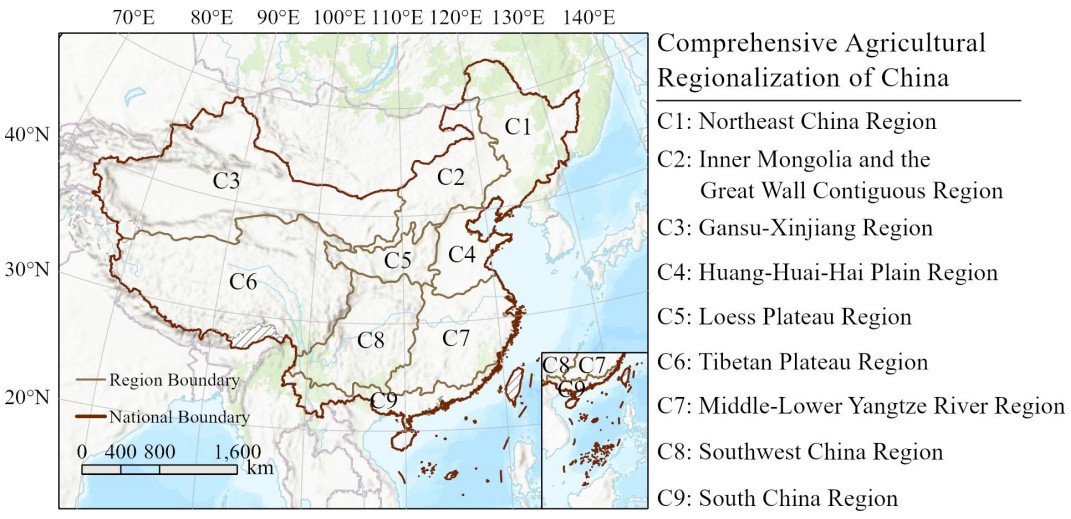

Fig.1 Agricultural Comprehensive Zoning. This map was created using Esri's ArcGIS® software. ArcGIS® and ArcMap™ are proprietary trademarks of Esri, used herein under license. © Esri. All rights reserved. For more information about Esri® software, please visit www.esri.com.

## 2.2 Data

We utilized soil moisture data, precipitation data, and phenological data of the three major cereal crops. The soil moisture data were standardized to identify drought events; the 99th percentile of precipitation data for each grid cell with rainfall was set as the extreme rainfall threshold; and the phenological data of the crops were used to calculate the exposure during the maturity stage. The specific data details are as follows:

(1) China 1km Soil Moisture Daily Dataset (2000-2022) based on station observations. National Tibetan Plateau Data Center. (https://cstr.cn/18406.11.Terre.tpdc.272415.)

(2) China Daily Precipitation Dataset (1961-2022): China Daily Precipitation Dataset (1961-2022, 0.1°/0.25°/0.5°): National Tibetan Plateau Data Center. (https://doi.org/10.11888/Atmos.tpdc.300523.https://cstr.cn/18406.11.Atmos.tpdc.300 523.)

(3) National Three Major Grain Crops 1km Planting Distribution Dataset (2000-2019): Luo Yuchuan; Zhang Zhao. National Three Major Grain Crops 1km Planting Distribution Dataset [DS/OL]. National Ecological Science Data Center.

(https://doi.org/10.12199/nesdc.ecodb.rs.2022.016.https://cstr.cn/15732.11.nesdc.ecodb.rs.20 22.016.)

**Table 1 Data Sources**

| Data Name | Time Span | Spatial resolution | Time resolution | Source |
|---|---|---|---|---|
| Soil Moisture | 2000-2022 | 1km×1km | Daily | National Tibetan Plateau Data Center |
| Precipitation | 1961-2022 | 9km×9km | Daily | National Tibetan Plateau Data Center |
| Crop Phenological Stages | 2000-2019 | 1km×1km | Daily | National Ecological Science Data Center |

## 2.3 Methodology

### 2.3.1 Identification of CDER

Compound events in this study are divided into two types: one is extreme rainfall

followed by drought within 5 days ($\text{CDER}_{rd}$), and the other is drought followed by extreme
rainfall within 5 days ($\text{CDER}_{dr}$) (Sun et al., 2024). The intensity of compound drought and
extreme rainfall events is composed of three parts: drought intensity, extreme rainfall
intensity, and the interval time between the two events (**Eq. 1**). Extreme rainfall is defined as
159  days with rainfall exceeding the 99th percentile threshold of the grid's rainfall days
(Schillerberg and Tian, 2024). A drought event is defined as a day on which the standardized
soil moisture index (SSMI) in the region falls below one negative standard deviation of the
21 years mean value of this index. For drought events, this study first applies a moving
average with a window size of 7 to smooth the data. Subsequently, soil moisture data for the
same period in historical records are standardized, and values below -σ were identified as
drought events. The absolute value of these standardized anomalies is defined as drought
intensity. For extreme rainfall, all extreme rainfall events within the same region are
standardized and uniformly shifted to be greater than zero. The resulting values are used to
represent extreme rainfall intensity. Drought events are identified using the standardized soil
moisture index, with the average value during the entire drought duration representing
drought intensity.

$$C = \frac{P \times D}{\Delta t} \tag{1}$$

Here $C$ represents the compound event intensity, $P$ represents the extreme rainfall
intensity, $D$ represents the drought intensity, and $\Delta t$ represents the interval time between
extreme rainfall and drought events.

### 2.3.2 Mechanism of Soil Damage from CDER

Compound events involving drought and extreme rainfall exert synergistic negative
impacts on soil health and agricultural productivity. Extreme rainfall events induce severe
soil erosion, resulting in the loss of fine particles and essential nutrients. Simultaneously,
prolonged drought conditions accelerate the decomposition and depletion of soil organic
matter (SOM), further weakening soil structure. Together, these processes significantly
increase soil erosion susceptibility. Drought-induced degradation of soil aggregate stability
reduces porosity and water retention capacity. When followed by intense, short-duration
rainfall, the already-compromised soil structure is further damaged by surface runoff and the

destruction of soil pores, leading to a sharp decline in the infiltration rate. As a result, the
limited water that is delivered during extreme rainfall events fails to effectively rehydrate the
soil, compounding the water deficit stress experienced by crops and impairing agricultural
resilience.

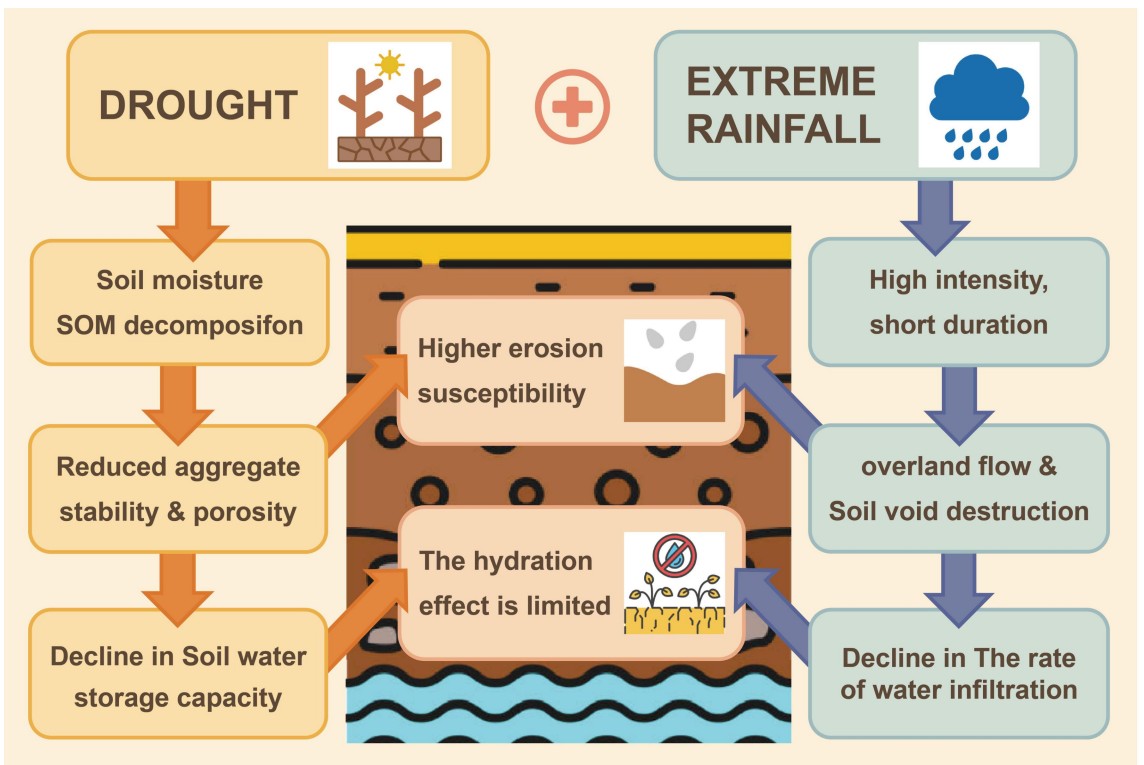

Fig.2 Schematic Diagram of Mechanism.

### 2.3.3 Calculation of Crops Exposure to CDER

The exposure during the growth stage of each crop is calculated by multiplying the
number of compound events occurring within the growth stage of each grid by the
agricultural land area of that grid (Eq. 2). We use the crop maturity date in combination with
the typical maturity period length of the three major crops to backtrack and determine the
time window for the entire growing season, within which we count the occurrences of
compound events to obtain $f$. Specifically, we select 130 days for maize, 100 days for rice,
300 days for winter wheat, and 100 days for spring wheat. In China, a large proportion of rice
cultivation consists of double-cropping rice, which includes early rice and late rice. Due to its
higher yield, better grain quality, greater economic value, and increased vulnerability to
CDER, this study focuses exclusively on late rice.

The distribution of different varieties of the three major cereal crops across China is relatively extensive. Therefore, it is not feasible to represent the growth period of each crop type by selecting a single representative variety; instead, multiple data sources need to be collected and weighed comprehensively.

For rice, the average growth period of the late rice in double-cropping systems used in this study is approximately 88 days (Liu et al., 2018), and around 117 days (Li et al., 2011). Ultimately, this study adopts a growth period of 100 days for rice.

For maize, the major varieties cultivated in China and their respective growth periods investigated in this study are as follows: Xianyu 335 (Zea mays L. cv. Xianyu335), approximately 130 days; Demeiya 1 (Zea mays L. cv. Demeiya1), approximately 110 days; Denghai 605 (Zea mays L. cv. Denghai605), ranging from 100 to 130 days; Longping 206 (Zea mays L. cv. Longping206), approximately 101 days; Jindan 1771 (Zea mays L. cv. Jindan1771), approximately 125 days; Gaonong 1206 (Zea mays L. cv. Gaonong1206), approximately 129 days; Shanyu 580 (Zea mays L. cv. Shanyu580), approximately 130 days; Tianyu 219 (Zea mays L. cv. Tianyu219), approximately 129 days. Taking these into account, this study uses 130 days as the growth period for maize.

Regarding spring wheat, the book *Chinese Wheat Cultivation* (Jin, 1961) indicates that the growth period of spring wheat generally exceeds 100 days, with the shortest around 80 days and the longest reaching approximately 190 days. The growth period of winter wheat typically exceeds 240 days, and can reach over 350 days in regions such as Linzhi, Tibet. Therefore, this study adopts a growth period of 100 days for spring wheat and 300 days for winter wheat.

$$Exp_{agr} = Agr \times f \qquad (2)$$

Where $f$ represents the frequency of compound events, $Exp_{agr}$ represents the agricultural exposure, and Agr represents the agricultural land area.

### 2.3.4 Other Statistical Methods

This study employed several methods to analyze CDER. Soil moisture data are standardized to identify drought conditions by transforming them to a distribution with a mean of 0 and a standard deviation of 1. The Least Squares Method (LSM) is used to fit a

linear trend and assess annual changes in the frequency of compound events. The K-Means clustering algorithm classifies event intensity into five levels based on data similarity. Loess regression is applied to model local trends in crop risk evolution, capturing non-linear patterns through weighted least squares within local neighborhoods. These approaches enabled a comprehensive analysis of extreme climate events in relation to crop phenology and soil moisture.

## 3.Result

### 3.1 Spatiotemporal Characteristics of CDER

This study systematically analyzed the spatial distribution characteristics of compound drought-rainfall events ( $CDER_{dr}$ ) and compound rainfall-drought events ( $CDER_{rd}$ ) across mainland China (**Fig 3A, 3C**), and visualized their disaster intensities using a five-level intensity classification system (**Fig 3B, 3D**). The results indicate: The annual occurrence frequency of $CDER_{dr}$ events is significantly higher than that of $CDER_{rd}$ events, with high-frequency aggregation zones formed in the Northwest Arid Region, Hengduan Mountains, and central-northern Xinjiang. Moreover, $CDER_{dr}$ events exhibit spatial heterogeneity characteristics in occurrence frequency compared to $CDER_{rd}$ events in northern regions. Although no significant differences were observed in intensity magnitudes between the two event types, high-intensity $CDER_{rd}$ zones (Level IV-V) are concentrated in northeastern China and northeastern Inner Mongolia (110°-113°E, 37°-40°N), while $CDER_{dr}$ events demonstrate spatially balanced intensity distribution with their high-intensity zones (Level III-IV) primarily forming a belt along the hilly belt of southern China (23°-25°N).

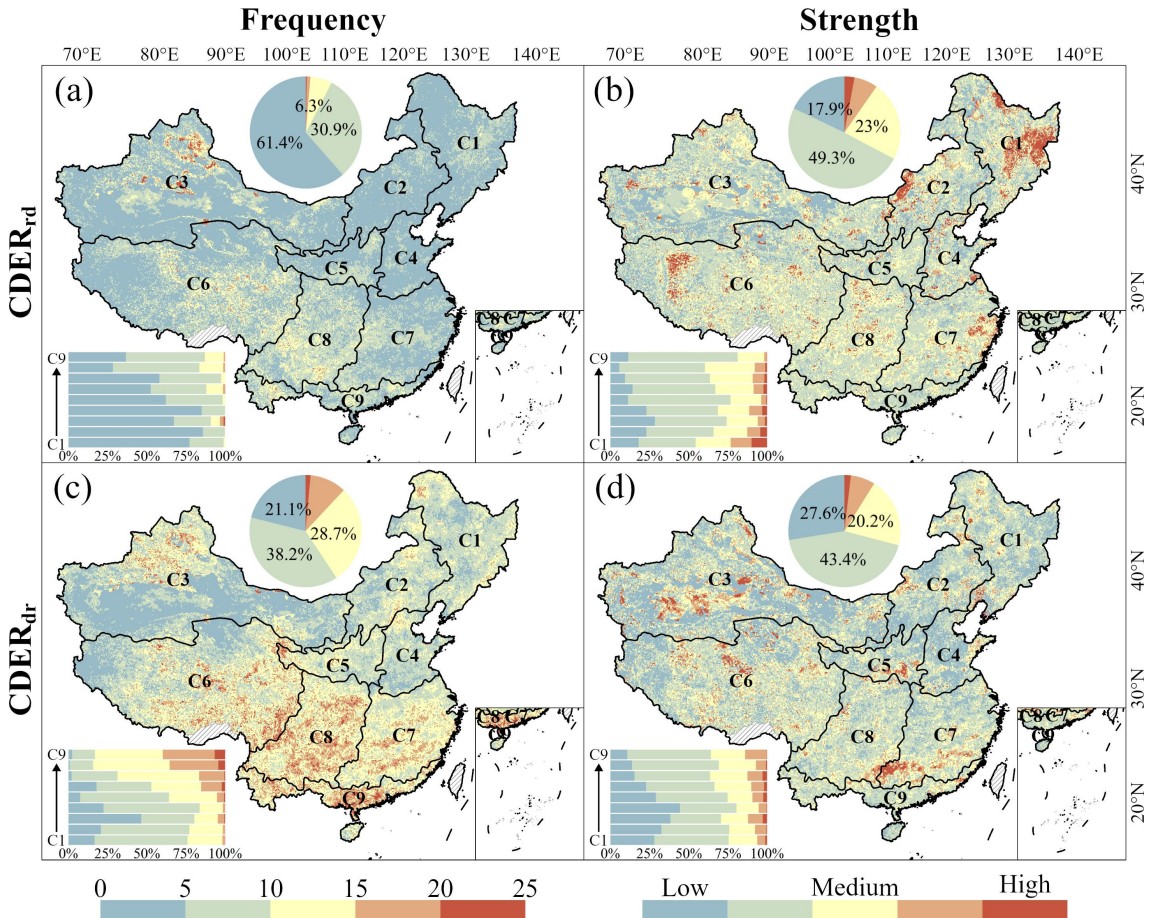

Fig.3 Spatial Distribution of Compound Drought and Extreme Rainfall Events. (a) and (b) show the frequency and intensity of $CDER_{rd}$ respectively, while (c) and (d) depict the frequency and intensity of $CDER_{dr}$. The bar charts represent the proportion of different color values within each region, while the pie charts show the proportion of each magnitude across the entire country.

There are distinct seasonal differences between $CDER_{dr}$ and $CDER_{rd}$. This study statistically analyzes the monthly average proportions of affected areas for $CDER_{rd}$ and $CDER_{dr}$ (**Figs. 4**). Among the nine major agricultural regions, the temporal variation trends in affected area proportions across different regions are similar, with no significant inter-regional differences. The bar charts for each region exhibit unimodal distributions: northern regions (C1~C6) reach peak coverage in July, while southwestern and southern regions (C7~C8) peak in June when $CDER_{dr}$ achieve maximum coverage. Overall, $CDER_{dr}$ show the highest disaster coverage during summer months (June, July, August),

 significantly exceeding other seasons, followed by spring and autumn. Across all
agricultural regions, the affected area proportions of $CDER_{dr}$ from December to February
are nearly zero, indicating extremely low winter occurrence frequencies, with limited
distributions only in the Middle-Lower Yangtze River region, southwestern region, and
southern region.

Simultaneously, comparing monthly affected area proportions between $CDER_{dr}$ and
$CDER_{rd}$ reveals consistent inter-monthly variation trends: similar spatial distributions
concentrated in summer, secondary peaks in spring/autumn, and minimal winter
occurrences. However, the affected area proportions of $CDER_{dr}$ are significantly higher
than those of $CDER_{rd}$, nearly doubling the latter. For instance, in region C1, the July
coverage proportion of $CDER_{dr}$ reaches 20%, whereas $CDER_{rd}$ peak at only ~10%. Overall,
$CDER_{dr}$ coverage proportions are approximately twice those of $CDER_{rd}$. Notably, in
region C5, the August affected area proportion of $CDER_{dr}$ exceeds that in September,
whereas $CDER_{rd}$ show the opposite pattern.

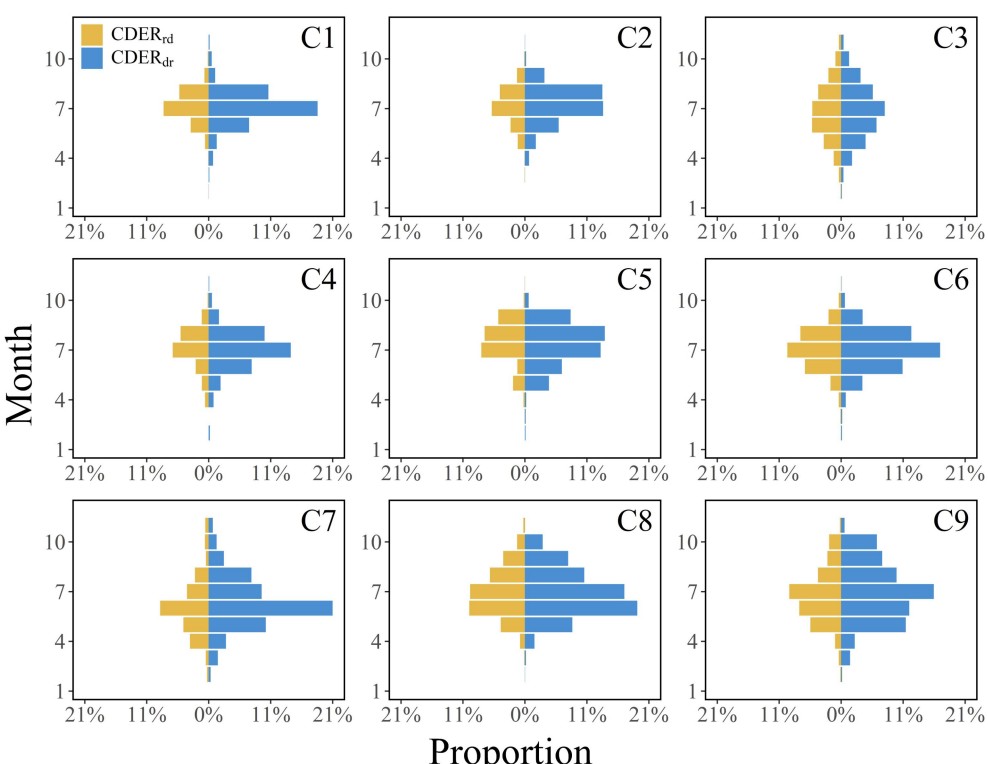

Fig.4 Monthly Average Affected Area of Compound Drought-Rainfall Events. C1 to

C9 represent the nine major agricultural regions mentioned earlier, with yellow indicating $CDER_{rd}$ and blue representing $CDER_{dr}$.

The interannual variation trends in affected areas of the two compound event types exhibit regional differences. This study further analyzes the changing trends in affected area proportions of $CDER_{dr}$ and $CDER_{rd}$ across agricultural regions from 2000 to 2020 (**Figs. 5, 6**). The nine agricultural regions show similar distribution patterns, with most regions maintaining relatively stable interannual variation trends and insignificant changes. Regional heterogeneity primarily manifests in C1, C3, and C6~C9, where both compound types exhibit declining trends ($p<0$), with $CDER_{dr}$ showing more pronounced frequency reductions. Specifically, C1 and C3 demonstrate insignificant downward trends characterized by relatively stable event frequencies with minor fluctuations. In C3, annual mean frequencies of $CDER_{dr}$ and $CDER_{rd}$ generally exhibit synchronized fluctuations, whereas no significant correlation is observed between them in C1. The declining trends in C6~C9 are notably stronger, collectively showing higher frequencies before 2010 followed by significant post-2010 reductions, despite occasional minor rebounds.

C2 and C4 display upward trends, with no apparent correlation between annual mean frequencies of $CDER_{dr}$ and $CDER_{rd}$. Notably, although C4 shows an overall increasing trend, it exhibits consecutive declines during 2018-2020, suggesting potential future frequency reductions. The Loess Plateau region (C5) presents a unique case: $CDER_{dr}$ frequencies show a declining trend while $CDER_{rd}$ frequencies exhibit an upward trend, accompanied by substantial overall fluctuations and relatively poor model fitting performance.

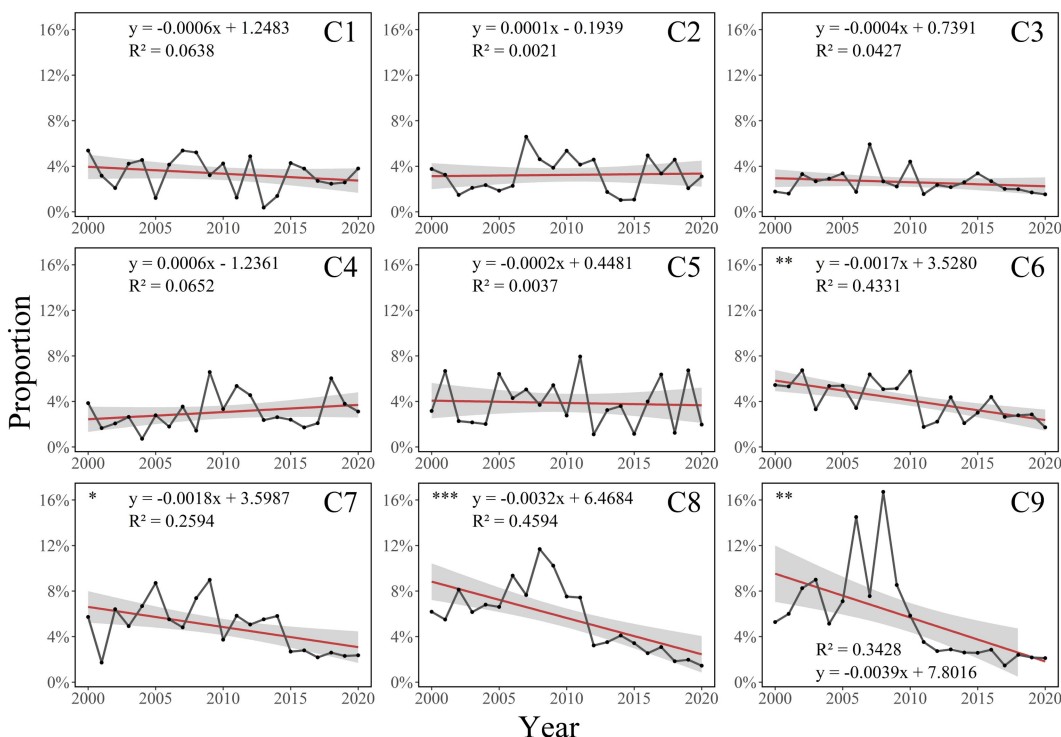

Fig. 5 Annual Frequency of Compound Drought-Rainfall Events. Red line represents the fitted trend line, and the shaded area represents the confidence interval. One star indicates p < 0.05, two stars indicate p < 0.01, and three stars indicate p < 0.001.

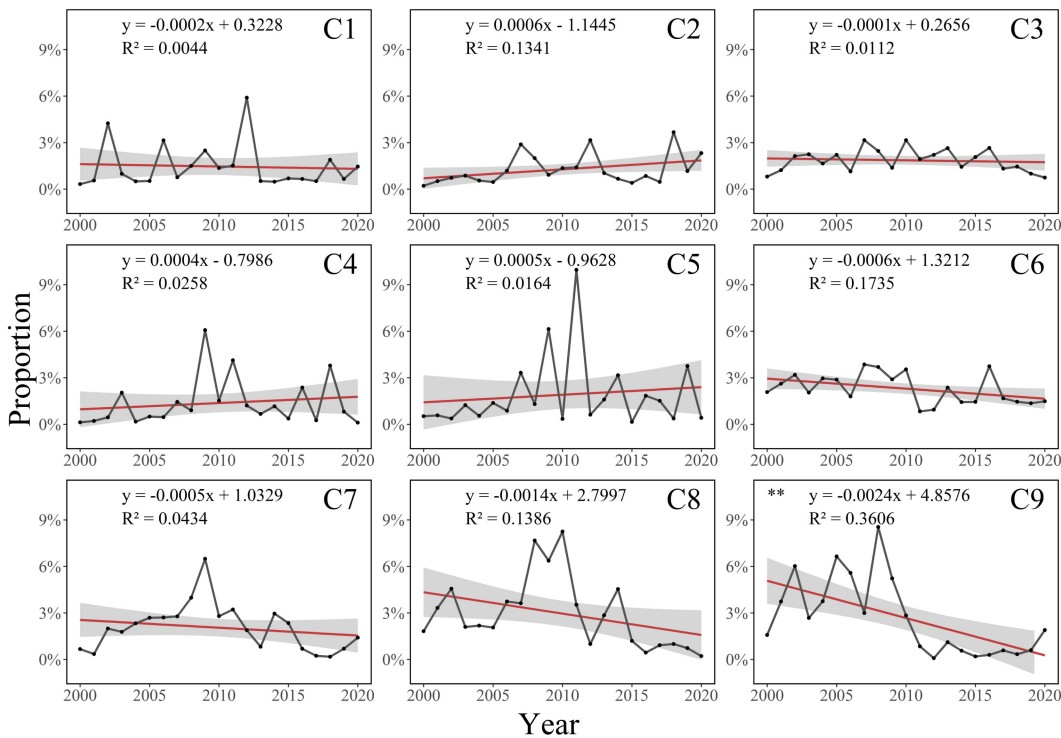

Fig. 6 Annual Frequency of Compound Rainfall-Drought Events. Red line represents the fitted trend line, and the shaded area represents the confidence interval. One star indicates p < 0.05, two stars indicate p < 0.01, and three stars indicate p < 0.00.

**3.2 Agricultural Production Exposure Analysis**

3.2.1 Spatial Distribution of Compound Event Risk in Nine Major Agricultural Regions

By overlaying $CDER_{rd}$ and $CDER_{dr}$ across China's nine major agricultural regions, this study generates boxplots illustrating exposure for three primary crops (maize, rice, wheat) in these regions (**Fig. 7**). The boxplots reveal significant regional disparities in disaster risks among maize, rice, and wheat, reflecting notable fluctuations in crop exposure risks across regions. Overall, C8 demonstrates universally higher disaster risks for all crops, while geographically distinct regions like C1, C2,C3 and C6 exhibit lower risks. The exposure risk of maize and wheat covers a broader national extent, being widely distributed across all regions. Region C4 exhibits exposure risk that is markedly higher than in any other region, followed by C5 and C8, where the risk is chiefly

concentrated on the North China Plain. A notable distinction between the two cereals is that maize exposure risk is comparatively elevated in C8, whereas wheat exposure risk in the same region remains relatively low. Owing to its geographically restricted cultivation zone, rice shows exposure risk only in C4, C7, C8, and C9; among these, region C7 registers the highest risk, and the areas of heightened risk are mainly situated in the middle reaches of the Yangtze River and the Pearl River Basin.

Among the three crops, wheat shows the highest average national risk, followed by maize, while rice displays minimal exposure risks due to limited cultivation areas nationwide. Maize exhibits generally higher and more variable risks, particularly in C4 and C5, likely attributable to extensive cultivation and frequent disasters in these regions. Wheat demonstrates relatively lower and more stable risks, with concentrated vulnerabilities in C4 and C5. Rice maintains the lowest risks with minimal fluctuations, potentially linked to its climatic adaptability. Although C7, C8, and C9 show elevated rice risks, limited fluctuations suggest manageable cultivation risks.

Regionally, most agricultural zones display clear correlations between crop exposure risks and cultivation scales. Both C1 and C6 exhibit low risks with minor fluctuations for all crops, though for divergent reasons: favorable climate and rare disasters in the Northeast (C1) versus low cultivation intensity in the Qinghai-Tibet Plateau (C6). Notably, C8 shows moderate-to-high across all crops, indicating combined impacts of local climate and compound disasters on cultivation.

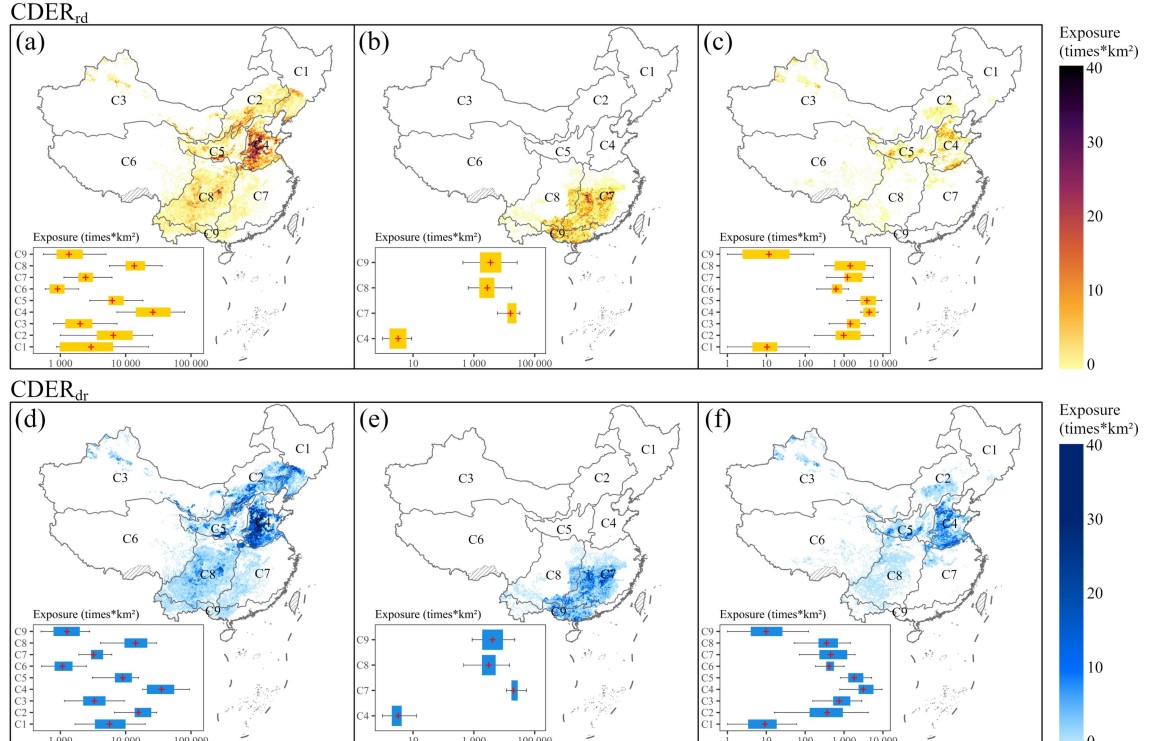

CDER$_{rd}$

CDER$_{dr}$

Fig. 7 Boxplots and Spatial Distribution Maps of Exposure Risk for the Three Major Crops in the Nine Agricultural Regions. (a), (b), (c) represent the exposure during CDER$_{rd}$ for maize, rice, and wheat, respectively; (d), (e), (f) represent the exposure during CDER$_{dr}$ for maize, rice, and wheat,The box plots display the distribution of exposure values across the different regions.

3.2.2 Annual Changes in exposure of the Three Major Crops

Based on maize exposure data from 2000-2019 across China's nine major agricultural regions, fitting curves were constructed (**Fig. 7**). Regionally, C4 exhibits the highest exposure (up to 60,000), followed by C1, C2, C5, and C8 (reaching ~ 10,000), with lower exposure in other regions. Comparatively, CDER$_{dr}$ generally demonstrates higher exposure than CDER$_{rd}$, though exceptions occur: CDER$_{rd}$ risks exceed CDER$_{dr}$ in C8 around 2008 and in C9 during 2011-2020.Trend analysis reveals: Most regions show similar exposure trajectories for both compound types. C1, C3, C4, and C7 display initial increases followed by declines and subsequent rises, peaking around 2010, troughing near 2016, then rebounding. C6, C8, and C9 exhibit rise-fall-rise-decline patterns with varying

peak/trough years. C2 and C5 demonstrate distinct behaviors: C2 shows rise-decline trends for both compound types, peaking around 2011 before decreasing; C5 displays rise-decline trends under $CDER_{dr}$ but rise-decline-rise patterns under $CDER_{rd}$ . Simultaneously, It is noteworthy that the exposure of C3, C6, and C9 rose in lockstep with the expansion of maize-cultivated area. By contrast, although the cultivated areas of C1, C2, C4, C5, C7, and C8 increased steadily from 2000 to 2019, their exposure did not exhibit a sustained upward trajectory; rather, it generally peaked around 2009 and subsequently declined, revealing marked differences in trend. These patterns indicate that maize exposure is governed chiefly by intrinsic factors — such as the frequency of hazardous events — while the influence of cultivated area, though present, is comparatively modest.

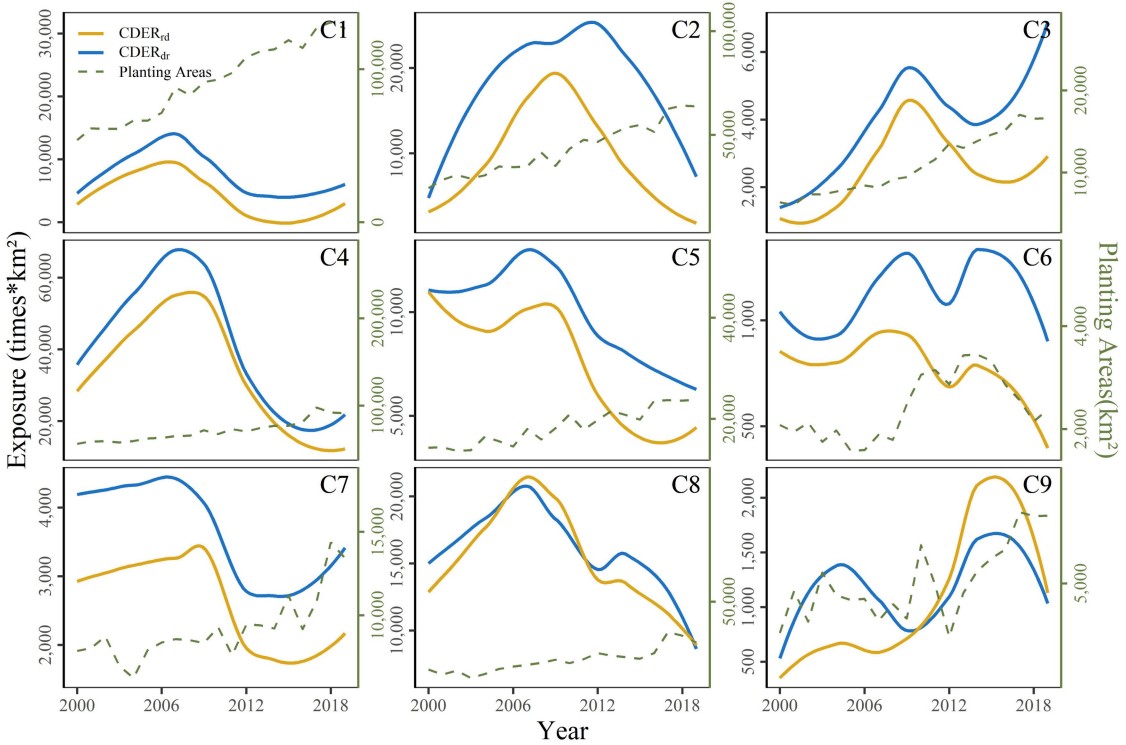

Fig. 8 Fitted Curve of Maize Exposure Risk During $CDER_{rd}$ in the Nine Agricultural Regions (2000-2019). The yellow curve represents the annual exposure trend for $CDER_{dr}$, the blue curve represents the annual exposure trend for $CDER_{rd}$, while the green line shows the annual variation in planting area.

Based on rice exposure data from 2000-2019 across China's nine major agricultural regions, fitting curves were constructed (**Fig. 9**). Regions C1 (Northeast), C2 (Inner Mongolia-Great Wall), C3 (Gansu-Xinjiang), C5 (Loess Plateau), and C6 (Qinghai-Tibet Plateau) exhibit zero exposure throughout the period due to negligible rice cultivation and are excluded from presentation. Regionally, C7 shows the highest exposure (up to 40,000), followed by C8 and C9, with C4 displaying the lowest. Comparatively, exposure between $CDER_{dr}$ and $CDER_{rd}$ are generally similar, though $CDER_{rd}$ risks were lower than $CDER_{dr}$ prior to 2013 but higher post-2013. However, in C7, $CDER_{rd}$ exposure remain consistently lower than $CDER_{dr}$. Trend analysis reveals divergent evolutionary patterns: C4 demonstrates an initial rise followed by decline and subsequent increase, peaking around 2006 before declining sharply. C7 and C8 both exhibit rise-fall-rise trajectories, peaking around 2010, troughing near 2012, then rebounding. Southern China (C9) displays distinct behaviors: $CDER_{dr}$ exposure follows a rise-decline pattern, while $CDER_{rd}$ exposure shows rise-decline-rise-decline fluctuations. Meanwhile, it is noteworthy that the exposure of C7, C8, and C9 changes in lockstep with the rice-planted area. In particular, although the rice-planted area in C4 has increased, its exposure has shown a consistently declining trend, revealing distinct differences among the trajectories. These findings indicate that rice exposure is strongly influenced by intrinsic factors such as event frequency, yet the planted area also exerts a substantial effect. This may be attributable to the relatively stable frequency and intensity of events in rice-growing regions, which accentuates the impact of changes in planted area.

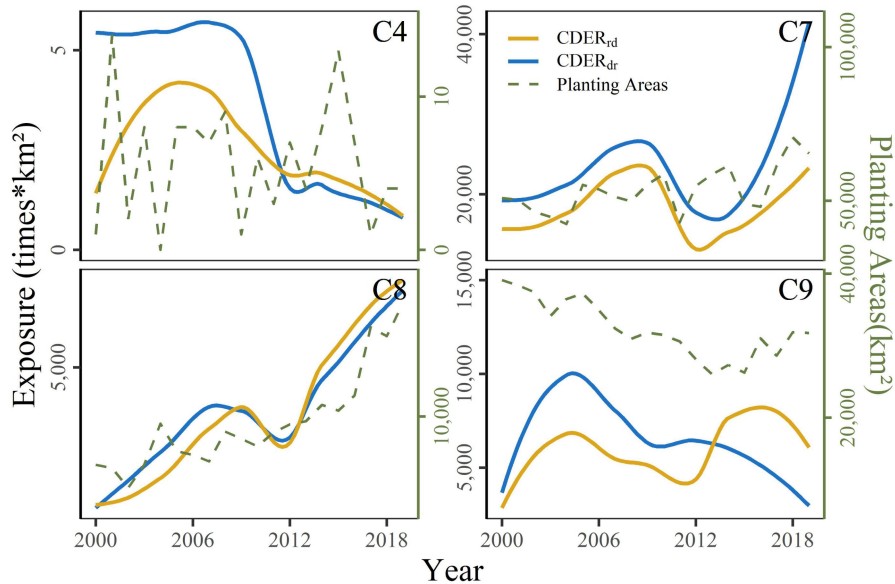

Fig. 9 Fitted Curve of Rice Exposure Risk During $CDER_{rd}$ in the Nine Agricultural Regions (2000-2019). The yellow curve represents the annual exposure trend for $CDER_{dr}$, the blue curve represents the annual exposure trend for $CDER_{rd}$, while the green line shows the annual variation in planting area.

Fig. 10 presents the fitting curves derived from wheat exposure data (2000-2019) across China's nine major agricultural regions. Regionally, C4 ranks first with a exposure index of 30,000, followed by C5 at 10,000, while other regions generally exhibit risks below 5,000. The high-risk characteristics of C4 and C5, as China's core wheat production zones, show significant correlations with regional cultivation scales. Comparatively, $CDER_{dr}$ -induced exposure are nearly double those of $CDER_{rd}$ . Trend analysis reveals substantial differences between the two compound types: $CDER_{dr}$ exposure universally follow rise-decline patterns with peak years clustered around 2010, though specific timing varies. Conversely, $CDER_{rd}$ trends exhibit greater regional heterogeneity: C3, C5, C6, and C9 display decline-rise-decline-rise patterns (troughing around 2004, peaking near 2010, troughing again around 2014, then rebounding); C4, C7, and C8 show sustained declines , C7 experienced a brief rebound in 2013 before resuming its decline, whereas C4 rebounded after 2016 and has continued to climb since then; C2 demonstrates unique rise-decline-rise-decline fluctuations, peaking around 2004, troughing near 2007, rebounding to 2012 highs, then declining. At the same time, the

relationship between wheat exposure and its planted area proves to be relatively complex. Specifically, the exposure of C3, C8, and C9 increases in tandem with expansions in planted area, whereas in C2, C4, C5, C6, and C7 the planted area remains essentially stable while exposure fluctuates markedly, showing no comparable trend. Particularly noteworthy is C1, where the planted area varies substantially, yet exposure has consistently remained at an extremely low level. These findings suggest that wheat exposure is governed primarily by intrinsic factors—such as the frequency of hazardous events—while planted area, though influential to some extent, plays a comparatively minor role.

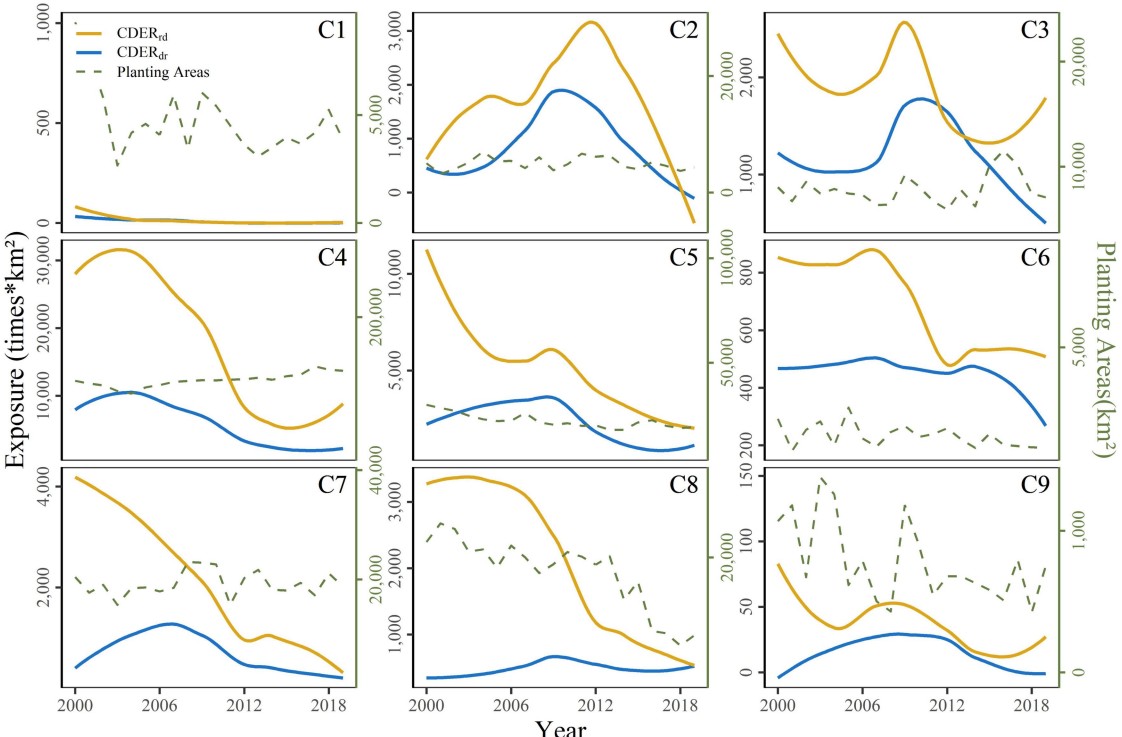

Fig. 10 Fitted Curve of Wheat Exposure Risk During $CDER_{rd}$ in the Nine Agricultural Regions (2000-2019). The yellow curve represents the annual exposure trend for $CDER_{dr}$, the blue curve represents the annual exposure trend for $CDER_{rd}$, while the green line shows the annual variation in planting area of Spring wheat and winter wheat.

For maize and rice, $CDER_{dr}$ -induced exposure is significantly higher than that from $CDER_{rd}$, whereas for wheat, $CDER_{dr}$ exposure is significantly lower than that from $CDER_{rd}$. However, the double proportional relationship between exposure risks and affected areas of

the two compound event types does not universally hold. The study reveals that agricultural

regions with larger cultivation areas exhibit exposure ratios between $CDER_{dr}$ and $CDER_{rd}$

closer to the double proportion of their respective affected areas. This phenomenon stems

from spatial non-overlap between compound disaster zones and major crop cultivation areas.

As agricultural cultivation expands, spatial coupling between disaster-affected zones and

croplands intensifies, driving progressive increases in exposure. Notably, green polylines

reflect interannual variations in crop cultivation areas across regions. Combined analysis of

cultivation area trends and exposure calculations reveals two primary scenarios: (1) Inverse

trends—exposure decrease despite cultivation area expansions, or increase despite area

reductions; (2) Asymmetric fluctuations—stable cultivation areas with volatile exposure risks,

or stable exposure with fluctuating cultivation areas. Both scenarios demonstrate that

interannual exposure variations primarily result from intrinsic changes in $CDER_{dr}$

characteristics rather than cultivation area adjustments.

## 4. Discussion

### 4.1 Consistency Analysis of the Spatiotemporal Characteristics of CDER

Numerous studies have focused on the distribution patterns and impacts of drought-

flood abrupt alternation events. These events share similarities with the $CDER_{dr}$ studied here,

but they are typically approached from a hydrological perspective using runoff data, whereas

our study focuses on the agricultural system using soil moisture data. Moderate rainfall

following drought generally has a positive impact on agricultural production and the

ecological environment. However, if drought abruptly turns into flooding, it can exacerbate

soil erosion and other disasters, causing more severe impacts on crops and worsening water

quality (Bi, 2022; Huang et al., 2019; Shi et al., 2022). In the field of $CDER_{rd}$ and $CDER_{dr}$,

previous studies have indicated that in the Yangtze River middle and lower reaches, the

frequency of $CDER_{dr}$ is mainly concentrated in June each year, while the intensity shows

certain fluctuations (Shen et al., 2018). In the Jianghuai-Huai River Basin, the onset of the

rainy season in drought-prone and semi-humid regions is delayed by 1–2 months compared

to that in humid regions, making the semi-humid and humid areas high-incidence regions for

drought-flood abrupt alternation events (Xue et al., 2024). Additionally, studies in Fujian Province, China, have found that $CDER_{rd}$ occur more frequently in February, July, and August, while $CDER_{dr}$ have a higher occurrence rate from June to October (Zhang et al., 2018). These findings are highly consistent with our results, further validating the spatiotemporal distribution characteristics of compound events in these regions.

However, due to significant differences in the identification methods, definition criteria, and time scales used in different studies, some discrepancies inevitably exist in the results. Our study shows that $CDER_{dr}$ have significantly higher frequency and intensity than $CDER_{rd}$, with values approximately twice as high. Yet one study using the Standardized Precipitation Evapotranspiration Index (SPEI) as an indicator of compound events found that the frequency and intensity of drought-to-flood and flood-to-drought events were comparable (Qiao et al., 2022). Moreover, several scholars have explored the correlation between drought-flood abrupt alternation events and complex climate factors (Bian, 2023; Wang et al., 2024), but the specific physical mechanisms behind these events are still under investigation, with no unified conclusions yet formed.

**4.2 Uncertainty in the Impact of CDER on Crop Growth Stages**

This study focuses on the maturity months of the three major grain crops and uses the monthly frequency of $CDER_{rd}$ and $CDER_{dr}$, along with 1km spatial resolution crop planting data, to characterize the exposure scenarios of the three major grain crops during their maturation periods in China's nine agricultural regions. Previous studies on crop exposure have mostly used annual time scales, such as those calculating exposure of population and farmland on a global scale for historical and future periods (2005 and 2085) (Tabari and Willems, 2023). However, disaster occurrences throughout the year do not always coincide with agricultural activities, leading to overestimation of crop exposure. Our study, which refines the exposure assessment to the crop maturity period, overcomes this limitation and provides a more accurate exposure profile.

Studies have shown that drought stress significantly affects crop yield, with varying impacts on different growth stages and species. For rice, drought during the reproductive stage causes a greater yield reduction compared to the early stages (Boonjung and Fukai et al., 1996). Similarly, wheat shows continuous yield reductions throughout its growth cycle.

Maize is also more severely affected during the reproductive stage, with early-stage stress causing lasting damage to photosynthetic capacity (Daryanto et al., 2016; Ma et al., 2017). Overall, maize appears to be more sensitive to drought than wheat, with yield reductions of 39.3% and 20.6%, respectively, under 40% water reduction (Daryanto et al., 2016). Therefore, failing to consider the sensitivity of different growth stages of crops to compound drought and extreme rainfall events can lead to overestimation or underestimation of risks. We suggest that future research should focus on designing experiments or other forms of investigation to explore the sensitivity of different growth stages of the three major grain crops to compound disasters. Based on this, key growth stages should be identified to incorporate the vulnerability of the affected bodies into more refined exposure risk studies.

**4.3 Increase in CDER Events under Future Climate Change and Adaptation Strategies**

Analysis of our results indicates that CDER events have increased over the study period and are concentrated in the northwest, southwest and northern agricultural regions. This pattern reflects broader climate processes: a warmer atmosphere can hold more moisture, leading to more intense rainfall, while rising temperatures also heighten evaporation and soil drying. Such a combination intensifies the alternation between drought and flooding and exemplifies the "wet-gets-wetter, dry-gets-drier" trend. The IPCC warns that compound extremes are likely to become more frequent as the climate warms and that concurrent heatwaves and droughts will intensify. In East Asia, summer monsoon rainfall has become heavier and more variable in recent decades, and heavy rain along the Meiyu front is occurring more often due to changes in moisture convergence and circulation(Steensen et al., 2025).

Given these projections, adaptation strategies are essential. Remote-sensing research shows that combining soil-moisture and vegetation indices can provide agricultural drought warnings up to two months in advance, highlighting the need to expand soil-moisture monitoring and early warning systems in high-risk regions. Our study showed that maize and wheat face higher exposure risks than rice, underscoring the vulnerability of certain crops to these events. Improving early warning systems would help farmers and authorities prepare for abrupt shifts from drought to extreme rainfall. Complementary measures include breeding crop varieties tolerant to both drought and waterlogging, adopting precision agriculture and

efficient irrigation, and developing weather-index insurance and other risk-sharing instruments to spread losses. National and regional climate-adaptation plans should explicitly account for compound extremes, recognising that uncertainties remain in the physical mechanisms driving CDER and the sensitivity of different crop growth stages. Ongoing research into these mechanisms and the vulnerability of crops will be vital for designing robust strategies to maintain food security in a changing climate.

## 5. Conclusions

This study defines compound drought and extreme rainfall events, including $CDER_{dr}$ and $CDER_{rd}$, and analyzes their spatiotemporal distribution in China's nine major agricultural regions. High-intensity $CDER_{rd}$ are concentrated in the north, especially Northeast China and Inner Mongolia, while $CDER_{dr}$ are widespread in the northeast and south, particularly South China and the Jianghuai region. $CDER_{dr}$ occur with higher frequency and intensity, affecting up to 20% of the area, compared to 10% for $CDER_{rd}$. Both event types are most prevalent in summer, with regional differences observed in annual affected area changes, especially in the Northeast, Inner Mongolia, Great Wall, Gansu-Xinjiang, Huang-Huai-Hai, and Loess Plateau regions.

The study further refines the calculation of exposure to compound events for maize, rice, and wheat during their crop maturity periods. Results show significant regional differences in disaster risk, with C8 facing the highest risk for all crops, while regions like C1, C2, and C6 experience lower risks. Among the crops, wheat faces the highest risk, followed by maize, while rice has the lowest exposure due to limited planting areas. Meanwhile interannual exposure variations primarily result from intrinsic changes in characteristics rather than cultivation area adjustments. The risk evolution across regions follows a common pattern of rising and then declining, with a peak around 2010, coinciding with higher frequencies and intensities of compound events.

## Data Availability

(1) China 1km Soil Moisture Daily Dataset (2000-2022) based on station observations. National Tibetan Plateau Data Center. (https://cstr.cn/18406.11.Terre.tpdc.272415.)

(2) China Daily Precipitation Dataset (1961-2022): China Daily Precipitation Dataset (1961-2022, 0.1°/0.25°/0.5°): National Tibetan Plateau Data Center. (https://doi.org/10.11888/Atmos.tpdc.300523.https://cstr.cn/18406.11.Atmos.tpdc.300 523.)

(3) National Three Major Grain Crops 1km Planting Distribution Dataset (2000-2019): Luo Yuchuan; Zhang Zhao. National Three Major Grain Crops 1km Planting Distribution Dataset [DS/OL]. National Ecological Science Data Center.

(https://doi.org/10.12199/nesdc.ecodb.rs.2022.016.https://cstr.cn/15732.11.nesdc.ecodb.rs.20 22.016.)

## Author Contribution

**Hanming Cao:** Writing-original draft, Writing-review & editing, Conceptualization, Data curation, Investigation, Methodology, Validation. **Qiren Yang:** Writing-review & editing, Conceptualization, Visualization. **Wei Yang:** original draft & editing, Data curation, Methodology. **Lin Zhao:** Writing–review & editing, Funding acquisition, Project administration, Resources.

## Competing Interests

The contact author has declared that none of the authors has any competing interests.

## Acknowledgments

This research was funded by Third Xinjiang Scientific Expedition Program (2022xjkk0601), National Natural Science Foundation of China (42471085 and U22B2011), Natural Science Foundation of Hubei Province (2023AFB823).

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
