# Peer review of "China's Three Major Cereal Crops Exposure to Compound Drought and Extreme Rainfall Events"

_EGUsphere, 2025_

## Author Response (AR1)

**Response to Comments**

Dear Editor and Reviewers:

We sincerely thank you for your valuable comments, which have helped us improve the quality of our manuscript a lot. The original comments are shown in black, our responses are shown in red, and the corresponding revisions in the manuscript are highlighted in blue.

**Response to Anonymous Referee #1, 18 Jul 2025**

The manuscript provides a timely and valuable contribution by assessing the exposure of China's three major cereal crops to Compound Drought and Extreme Rainfall Events (CDER) from a crop growth-stage perspective. By integrating high-resolution spatial data with phenological information, the authors reveal the spatiotemporal patterns and crop-specific risks of CDER across nine key agricultural regions. The use of a dual-index method—combining standardized soil moisture and percentile-based rainfall thresholds—represents a methodological advancement, and the focus on dynamic exposure trends fills a critical gap in current agricultural disaster risk research. The findings have important implications for climate-resilient agricultural planning and disaster mitigation.Here are my questions and suggestions:

**Comment 1:** Keyword Correction: I noticed that the manuscript refers to "crop maturity exposure" as a keyword, whereas the study itself clearly focuses on the growth stage of crops when assessing exposure. This appears to be a terminology error and should be corrected for consistency and accuracy. Writing Format Details: A few stylistic and formatting issues should be addressed: On line 57, there is an unnecessary dash ("-") that should be removed. After the first appearance of "Compound Drought and Extreme Rainfall Events" and its acronym CDER, the full term should not be repeatedly spelled out. CDER should be used throughout the manuscript for conciseness and clarity. Justification of Growth Stage Lengths (Page 8, Line 181): The manuscript states that "we select 130 days for maize, 100 days for rice,

300 days for winter wheat, and 100 days for spring wheat." This assignment of growth stage lengths lacks explanation. The authors should provide a rationale or cite appropriate agronomic references to justify these durations, as they play a critical role in exposure calculation.

Response to Comment 1:

(1) We have corrected the issues related to keyword details.We have fixed the spelling error at line 57 as you pointed out.

Author's changes in the manuscript:

The desiccation of soil organic matter (SOM) under drought stress leads to its accelerated decomposition, resulting in a significant reduction in SOM content often by 40-60% (Goebel et al., 2011; Yang and Liu, 2020).

(2) We have standardized the abbreviation format for "Compound Drought and Extreme Rainfall Events."

(3) We provide the following explanation regarding the selection of the crop growth periods:

[revised manuscript text omitted]

Jin, S.: Chinese Wheat Cultivation, China Agriculture Press, 1961. (in Chinese)

**Comment 2:** Figure 4 Legend Clarification: The figure title for Figure 4 should clearly indicate what the yellow and blue colors represent, as this is currently unclear to the reader.

Response to Comment 2:

We have added the relevant explanation.

Author's changes in the manuscript:

[Figure]

Fig.4 Monthly Average Affected Area of Compound Drought-Rainfall Events. C1 to C9 represent the nine major agricultural regions mentioned earlier, with yellow indicating $CDER_{rd}$ and blue representing $CDER_{dr}$.

**Comment 3:** Figure Formatting: In Figures 3 and 7, the subplots are labeled using uppercase letters (A, B, C, D). To maintain consistency and follow standard formatting conventions, I recommend converting these to lowercase letters with parentheses, i.e., (a), (b), (c), (d), etc., to avoid unnecessary confusion.

Response to Comment 2: We have revised the figure accordingly to address this graphical detail.

Author's changes in the manuscript:

[Figure]

Fig.3 Spatial Distribution of Compound Drought and Extreme Rainfall Events. (a) and (b) show the frequency and intensity of $CDER_{rd}$ respectively, while (c) and (d) depict the frequency and intensity of $CDER_{dr}$. The bar charts represent the proportion of different color values within each region, while the pie charts show the proportion of each magnitude across the entire country.

[Figure]

Fig. 7 Boxplots and Spatial Distribution Maps of Exposure Risk for the Three Major Crops in the Nine Agricultural Regions. (a), (b), (c) represent the exposure during $CDER_{rd}$ for maize, rice, and wheat, respectively; (d), (e), (f) represent the exposure during $CDER_{dr}$ for maize, rice, and wheat, The box plots display the distribution of exposure values across the different regions.

**Response to Anonymous Referee #2, 22 Jul 2025**

This study investigated the spatial-temporal variations of compound drought and extreme rainfall events in China, and quantified the exposure of three major cereal crops to these compound events. The assessment provides a significant contribution to the understanding of compound event risk given its dynamic and crop growth-stage perspective. Moreover, this study uses soil moisture data to monitor compound events from an agricultural perspective, which represents a methodological novelty. While, it can be further improved:

**Comment 1:** Section 2.3.1, it is not clear how the standardized soil moisture index (SSMI) was defined in this study. Also, how are the extreme rainfall events standardized?

Response to Comment 1:

For drought events, this study first applies a moving average with a window size of 7 to smooth the data. Subsequently, soil moisture data for the same period in historical records are standardized, and values below -σ were identified as drought events. The absolute value of these standardized anomalies is defined as drought intensity. For extreme rainfall, all extreme rainfall events within the same region are standardized and uniformly shifted to be greater than zero. The resulting values are used to represent extreme rainfall intensity.

Author's changes in the manuscript:

For drought events, this study first applies a moving average with a window size of 7 to smooth the data. Subsequently, soil moisture data for the same period in historical records are standardized, and values below $-\sigma$ were identified as drought events. The absolute value of these standardized anomalies is defined as drought intensity. For extreme rainfall, all extreme rainfall events within the same region are standardized and uniformly shifted to be greater than zero. The resulting values are used to represent extreme rainfall intensity. Drought events are identified using the standardized soil moisture index, with the average value during the entire drought duration representing drought intensity.

$$C = \frac{P \times D}{\Delta t} \qquad\qquad (1)$$

Here C represents the compound event intensity, P represents the extreme rainfall intensity, D represents the drought intensity, and $\Delta t$ represents the interval time between extreme rainfall and drought events.

**Comment 2:** Lines 249-254, the statements are not consistent with Fig.5 &6. For example: "$CDER_{dr}$ showed no significant trend in C7 and C8", while it has significant decreasing trend in Fig.5.

Response to Comment 2:

In the original statement, we should change "$CDER_{dr}$" to "$CDER_{rd}$" at this location. In addition, due to our oversight, some figures and tables in the manuscript were mistakenly uploaded using an earlier version in which drought events were

defined differently, resulting in inconsistencies in the conclusions. This issue has been corrected in the revised manuscript, and a comprehensive check has been conducted.

Author's changes in the manuscript:

The interannual variation trends in affected areas of the two compound event types exhibit regional differences. This study further analyzes the changing trends in affected area proportions of $CDER_{dr}$ and $CDER_{rd}$ across agricultural regions from 2000 to 2020 (Figs. 5, 6). The nine agricultural regions show similar distribution patterns, with most regions maintaining relatively stable interannual variation trends and insignificant changes. Regional heterogeneity primarily manifests in C1, C3, and C6~C9, where both compound types exhibit declining trends ($p<0$), with $CDER_{dr}$ showing more pronounced frequency reductions. Specifically, C1 and C3 demonstrate insignificant downward trends characterized by relatively stable event frequencies with minor fluctuations. In C3, annual mean frequencies of $CDER_{dr}$ and $CDER_{rd}$ generally exhibit synchronized fluctuations, whereas no significant correlation is observed between them in C1. The declining trends in C6~C9 are notably stronger, collectively showing higher frequencies before 2010 followed by significant post-2010 reductions, despite occasional minor rebounds.

**Comment 3:** Fig.7, the label for the subplot in the caption are not consistent with the figure ("A1, A2, A3, B1, B2, B3" V.S. "A, B, C, D, E, F").

Response to Comment 3:

We have revised the details of the figures as requested.

Author's changes in the manuscript:

[Figure]

Fig. 7 Boxplots and Spatial Distribution Maps of Exposure Risk for the Three Major Crops in the Nine Agricultural Regions. (a), (b), (c) represent the exposure during $CDER_{rd}$ for maize, rice, and wheat, respectively; (d), (e), (f) represent the exposure during $CDER_{dr}$ for maize, rice, and wheat,The box plots display the distribution of exposure values across the different regions.

**Comment 4:** Lines 380-382 and Lines 393-395, the reference for these studies should be provided.

Response to Comment 4:

We have added the relevant references.

Author's changes in the manuscript:

In the field of CDERrd and CDERdr, previous studies have indicated that in the Yangtze River middle and lower reaches, the frequency of CDERdr is mainly concentrated in June each year, while the intensity shows certain fluctuations (Shen et al., 2018)

However, due to significant differences in the identification methods, definition criteria, and time scales used in different studies, some discrepancies inevitably exist

in the results. Our study shows that CDERdr have significantly higher frequency and intensity than CDERrd, with values approximately twice as high. Yet one study using the Standardized Precipitation Evapotranspiration Index (SPEI) as an indicator of compound events found that the frequency and intensity of drought-to-flood and flood-to-drought events were comparable (Qiao et al., 2022).

**Comment 5:** The inconsistent variations of planting area and crop exposure in Fig. 8, 9, &10 needs more discussions and explanations.

Response to Comment 5:

Simultaneously, It is noteworthy that the exposure of C3, C6, and C9 rose in lockstep with the expansion of maize-cultivated area. By contrast, although the cultivated areas of C1, C2, C4, C5, C7, and C8 increased steadily from 2000 to 2019, their exposure did not exhibit a sustained upward trajectory; rather, it generally peaked around 2009 and subsequently declined, revealing marked differences in trend. These patterns indicate that maize exposure is governed chiefly by intrinsic factors—such as the frequency of hazardous events—while the influence of cultivated area, though present, is comparatively modest.

Author's changes in the manuscript:

This phenomenon stems from spatial non-overlap between compound disaster zones and major crop cultivation areas. As agricultural cultivation expands, spatial coupling between disaster-affected zones and croplands intensifies, driving

progressive increases in exposure. Notably, green polylines reflect interannual variations in crop cultivation areas across regions. Combined analysis of cultivation area trends and exposure calculations reveals two primary scenarios: (1) Inverse trends－exposure decrease despite cultivation area expansions, or increase despite area reductions; (2) Asymmetric fluctuations－stable cultivation areas with volatile exposure risks, or stable exposure with fluctuating cultivation areas. Both scenarios demonstrate that interannual exposure variations primarily result from intrinsic changes in CDERdr characteristics rather than cultivation area adjustments.

**Response to Anonymous Referee #3, 28 Jul 2025**

This is an interesting paper addressing a subject of significant agricultural importance. A basic criticism is that it is short of meteorological insight into the compound drought extreme rainfall events that can be highly damaging to agriculture.

**Comment 1:** To assist readers in their meteorological comprehension of this compound risk, it would be instructive if the authors could provide some salient historical examples of drought/extreme rainfall, and the specific impact on rice, wheat and maize. In particular, the geographical extent of crop impact should be detailed.

Response to Comment 1:

In 2011, China's middle-lower Yangtze River basin experienced an abrupt hydrological shift from prolonged drought to extreme flooding. After suffering the most severe drought in sixty years during winter and spring, the region was struck by four successive heavy rainfall events in June, resulting in catastrophic flooding. The most severe impacts occurred in Jiangxi Province, where approximately three million people were affected and agricultural losses reached 90,400 hectares (Deng and Chen, 2013). The most widely cultivated rice variety in the region is considered highly sensitive to extreme precipitation events. Over the past two decades, extreme rainfall has reduced China's rice production by one-twelfth (Fu et al., 2023).

References

Deng, Y. and Chen, X.: Effects of Drought floods Abrupt Alternation on Growing Development of Rice and Consideration for Related Issues, Biol. Disaster Sci., 36, 217–222, 2013.

[revised manuscript text omitted]

**Comment 3:** It would then be informative if some comment could be made on the increasing likelihood of such key events in the future changing climate.

Response to Comment 3:

Under the context of global climate change, the likelihood of increased CDER events in the future is extremely high. This assessment is primarily based on the following scientific understanding: global warming enhances atmospheric water-holding capacity, making extreme rainfall events more frequent, while rising

temperatures intensify soil moisture evaporation, making droughts more severe, forming a polarization trend of "wet gets wetter, dry gets drier"; the spatiotemporal heterogeneity of precipitation in China's monsoon climate zone is further exacerbated, and the instability of summer monsoons increases the possibility of alternating extreme events. Recent studies indicate that compound extreme events will significantly increase globally, with the occurrence rate of related events potentially increasing by 20-40% by mid-century (Future increase in European compound events where droughts end in heavy precipitation). In response to this trend, the following recommendations are proposed: improve CDER monitoring and early warning systems based on soil moisture, with priority coverage of high-risk areas such as northwest, southwest, and northern China; accelerate the breeding of crop varieties resistant to both drought and waterlogging dual stresses and promote precision agriculture technologies; innovate agricultural insurance and regional risk-sharing mechanisms; integrate compound extreme events into national climate adaptation strategies and formulate forward-looking policies to ensure long-term food security.

Author's changes in the manuscript:

Analysis of our results indicates that CDER events have increased over the study period and are concentrated in the northwest, southwest and northern agricultural regions. This pattern reflects broader climate processes: a warmer atmosphere can hold more moisture, leading to more intense rainfall, while rising temperatures also heighten evaporation and soil drying. Such a combination intensifies the alternation between drought and flooding and exemplifies the "wet‐gets‐wetter, dry‐gets‐drier" trend. The IPCC warns that compound extremes are likely to become more frequent as the climate warms and that concurrent heatwaves and droughts will intensify. In East Asia, summer monsoon rainfall has become heavier and more variable in recent decades, and heavy rain along the Meiyu front is occurring more often due to changes in moisture convergence and circulation(Steensen et al., 2025).

Given these projections, adaptation strategies are essential. Remote‐sensing research shows that combining soil‐moisture and vegetation indices can provide agricultural drought warnings up to two months in advance, highlighting the need to

expand soil‑ moisture monitoring and early warning systems in high‑ risk regions. Our study showed that maize and wheat face higher exposure risks than rice, underscoring the vulnerability of certain crops to these events. Improving early warning systems would help farmers and authorities prepare for abrupt shifts from drought to extreme rainfall. Complementary measures include breeding crop varieties tolerant to both drought and waterlogging, adopting precision agriculture and efficient irrigation, and developing weather‑ index insurance and other risk‑ sharing instruments to spread losses. National and regional climate‑ adaptation plans should explicitly account for compound extremes, recognising that uncertainties remain in the physical mechanisms driving CDER and the sensitivity of different crop growth stages. Ongoing research into these mechanisms and the vulnerability of crops will be vital for designing robust strategies to maintain food security in a changing climate.

References

Steensen, B. M., Myhre, G., Hodnebrog, Ø., and Fischer, E.: Future increase in European compound events where droughts end in heavy precipitation, Npj Clim. Atmospheric Sci., 8, 267, https://doi.org/10.1038/s41612-025-01139-0, 2025.

---

## Referee Report (RR1)

The authors have addressed all my previous concerns in a very thorough and comprehensive manner. I appreciate the detailed point-by-point responses and the corresponding revisions made in the manuscript. The revisions have significantly improved the quality and clarity of the work. In my view, the manuscript now meets the standards for publication, and I have no further objections.

---

## Author Response (AR2)

Dear Editor and Reviewer:

We sincerely thank you for your valuable comments, which have helped us improve the quality of our manuscript a lot. The original comments are shown in black, our responses are shown in red, and the corresponding revisions in the manuscript are highlighted in blue.

**Response to Anonymous Referee #2**

**Comment 1:** There are some technical errors with the equations in the manuscript.

Response to Comment 1: We have addressed two issues to enhance the rigor of the manuscript: the multiplication notation (replacing $\times$ with $\cdot$) in Equations (1) and (2), and potential dimensional ambiguity in Equation (1), respectively. Corresponding clarifications have been added (line 173).

Author's changes in the manuscript:

$$C = \frac{P \cdot D}{\Delta t} \tag{1}$$

Here $C$ represents the compound event intensity, $P$ represents the extreme rainfall intensity, $D$ represents the drought intensity, and $\Delta t$ represents the interval time between extreme rainfall and drought events. We have rendered both P and D dimensionless for the analysis.

$$Exp_{agr} = Agr \cdot f \tag{2}$$

Where $f$ represents the frequency of compound events, $Exp_{agr}$ represents the agricultural exposure, and Agr represents the agricultural land area.

**Comment 2:** It is better to use "Intensity" rather than "Strength" in Fig.3 to make it consistent with the text.

Response to Comment 1: The detailed issues identified in the figure have been corrected (line 251).

Author's changes in the manuscript:

[Figure]

Fig.3 Spatial Distribution of Compound Drought and Extreme Rainfall Events. (a) and (b) show the frequency and intensity of $CDER_{rd}$ respectively, while (c) and (d) depict the frequency and intensity of $CDER_{dr}$. The bar charts represent the proportion of different color values within each region, while the pie charts show the proportion of each magnitude across the entire country.